# StableFDG: Style and Attention Based Learning for Federated Domain Generalization

**Jungwuk Park**[*]
KAIST
savertm@kaist.ac.kr

**Dong-Jun Han**[*]
Purdue University
han762@purdue.edu

**Jinho Kim**
SK Hynix
jinho123.kim@sk.com

**Shiqiang Wang**
IBM Research
wangshiq@us.ibm.com

**Christopher G. Brinton**
Purdue University
cgb@purdue.edu

**Jaekyun Moon**
KAIST
jmoon@kaist.edu

## Abstract

Traditional federated learning (FL) algorithms operate under the assumption that the data distributions at training (source domains) and testing (target domain) are the same. The fact that domain shifts often occur in practice necessitates equipping FL methods with a domain generalization (DG) capability. However, existing DG algorithms face fundamental challenges in FL setups due to the lack of samples/domains in each client's local dataset. In this paper, we propose StableFDG, a *style and attention based learning strategy* for accomplishing federated domain generalization, introducing two key contributions. The first is style-based learning, which enables each client to explore novel styles beyond the original source domains in its local dataset, improving domain diversity based on the proposed style sharing, shifting, and exploration strategies. Our second contribution is an attention-based feature highlighter, which captures the similarities between the features of data samples in the same class, and emphasizes the important/common characteristics to better learn the domain-invariant characteristics of each class in data-poor FL scenarios. Experimental results show that StableFDG outperforms existing baselines on various DG benchmark datasets, demonstrating its efficacy.

## 1 Introduction

Federated learning (FL) has now become a key paradigm for training a machine learning model using local data of distributed clients [26, 19, 11]. Without directly sharing each client's data to the third party, FL enables the clients to construct a global model via collaboration. However, although FL has achieved remarkable success, the underlying assumption of previous works is that the data distributions during training and testing are the same. This assumption is not valid in various scenarios with domain shifts; for example, although the FL clients only have data samples that belong to the source domains (e.g., images on sunny days and rainy days), the trained global model should be also able to make reliable predictions for the unseen target domain (e.g., images on snowy days). Therefore, in practice, FL methods have to be equipped with a domain generalization (DG) capability.

Given the source domains in the training phase, DG aims to construct a model that has a generalization capability to predict well on an unseen target domain. Various DG methods have been proposed in a centralized setup [39, 21, 36, 38, 16, 15, 24, 23]. However, directly applying centralized DG schemes to FL can potentially restrict the model performance since each client has limited numbers of data samples and styles in its local dataset. The local models are unable to capture the domain-invariant characteristics due to lack of data/styles in individual clients.

---

[*]Authors contributed equally to this work.

37th Conference on Neural Information Processing Systems (NeurIPS 2023).

Although several researchers have recently focused on DG for FL [25, 2, 35, 28], they still do not directly handle the fundamental issues that arise from the lack of data and styles (which represent domains) in individual FL clients. These performance limitations become especially prevalent when federating complex DG datasets having large style shifts between domains or having backgrounds unrelated to the prediction of class, as we will see in Sec. 4. Despite the practical significance of federated DG, this field is still in an early stage of research and remains a great challenge.

**Contributions.** In this paper, we propose StableFDG, a *style and attention based learning strategy* tailored to federated domain generalization. StableFDG tackles the fundamental challenges in federated DG that arise due to the lack of data/styles in each FL client, with two novel characteristics:

- We first propose *style-based learning*, which exposes the model of each FL client to various styles beyond the source domains in its local dataset. Specifically, we (i) design a style-sharing method that can compensate for the missing styles in each client by sharing the style statistics with other clients; (ii) propose a style-shifting strategy that can select the best styles to be shifted to the new style to balance between the original and new styles; and (iii) develop style-exploration to further expose the model to a wider variety of styles by extrapolating the current styles. Based on these unique characteristics, our style-based learning handles the issue of the lack of styles in each FL client, significantly improving generalization capability.

- We also propose an *attention-based feature highlighter*, which enables the model to focus only on the important/essential parts of the features when making the prediction. Our key contribution here is to utilize an attention module to capture the similarities between the features of data samples in the same class (regardless of the domain), and emphasize the important/common characteristics to better learn the domain-invariant features. Especially in data-poor FL scenarios where models are prone to overfitting to small local datasets, our new strategy provides advantages for complicated DG tasks by removing background noises that are unrelated to class prediction and focusing on the important parts.

The two suggested schemes work in a complementary fashion, each providing one necessary component for federated DG: our style-based learning improves domain diversity, while the attention-based feature highlighter learns domain-invariant characteristics of each class. Experiments on various FL setups using DG benchmarks confirm the advantage of StableFDG over (i) the baselines that directly apply DG methods to FL and (ii) the baselines that are specifically designed for federated DG.

## 2 Related Works

**Federated learning.** FL enables multiple clients to train a shared global model or personalized models without directly sharing each client's data with the server or other clients. FedAvg [26] is a well-known early work that sparked interest in FL in the machine learning community. Since then, various FL strategies have been proposed to handle the communication burden issue [31, 9], data heterogeneity issue [20, 12], adversarial attack issue [34, 29, 7], and personalization issue [3, 18, 5]. However, existing FL methods do not have generalization capabilities to predict well on an arbitrary unseen domain. In other words, most prior FL methods are not able to handle the DG problem.

**Domain generalization.** DG is one of the emerging fields in the AI community due to its significance in practical applications. Existing DG strategies based on domain alignment [17, 24, 23], meta-learning [16, 4, 15, 37] and style/data-augmentation [39, 21, 36, 38] have shown great success in a centralized setup where the whole dataset is accessible during training. Recently, style-augmentation methods [39, 21, 36] including MixStyle [39] and DSU [21] are receiving considerable attention due to their high compatibility with various tasks and model architectures. However, although existing DG solutions work well in a centralized setup, they face challenges in FL scenarios; in data-poor FL setups, prior works achieve relatively low performance due to the lack of data samples or domains in each client, resulting in compromised generalization capabilities. The applications of meta-learning or domain alignment methods could be also limited when domain labels are not accessible in each client. Compared to these prior works focusing on a centralized DG setup, we develop a style and attention based DG strategy tailored to FL. The advantages of our methodology against these baselines are confirmed via experiments in Sec. 4.

**Federated domain generalization.** Only a few recent works [25, 2, 35, 28] have focused on the intersection of FL and DG. Based on the training set distributed across clients, the goal of these works is to construct a global model that is generalizable to an unseen target domain. In [2], the authors

proposed to share the style statistics across different clients that could be utilized during local updates. However, this method does not utilize the styles beyond the clients' domains and shows limited performance in specific datasets where the data samples are not well separated in the style space. Moreover, it increases the computation and memory costs for generating new images in each client using a pretrained style transfer. In our work, we handle these issues via style exploration and the attention-based feature highlighter to train the model with novel styles while capturing the important knowledge of each class. In [25], the authors proposed to exchange distribution information among clients based on Fourier transform, especially targeting image segmentation tasks for medical data. The authors of [35] proposed a strategy for federated DG based on the domain-invariant feature extractor and an ensemble of domain-specific classifiers. Two regularization losses are developed in [28] aiming to learn a simple representation of data during client-side local updates.

Although these prior works [25, 35, 28] improve DG performance, the authors do not directly handle the issues that arise from limited styles and data in each client. Compared to these works, we take an orthogonal approach based on style and attention based learning to effectively learn style-invariant features while capturing common knowledge of classes. Experimental results in Sec. 4 reveals that our scheme outperforms existing ideas tackling federated DG in practical data distribution setups.

## 3 Proposed StableFDG Algorithm

**Problem formulation.** We consider a FL setup with $N$ clients distributed over the network. Let $\mathcal{S}_n = \{(x_i^n, y_i^n)\}_{i=1}^{\rho_n}$ be the local dataset of the $n$-th client, which consists of $\rho_n$ pairs of data sample $x$ and the corresponding label $y$. Here, each client $n$ can have data samples from either a single or multiple source domains in its local dataset $\mathcal{S}_n$. Previous works on FL focus on constructing a global model that predicts well on the overall dataset $\mathcal{S} = \{\mathcal{S}_1, \mathcal{S}_2, \ldots, \mathcal{S}_N\}$ or personalized models that work well on individual local datasets $\mathcal{S}_n$. In contrast to the conventional FL setup, given the overall dataset (or source domains) $\mathcal{S}$, the goal of this work is to construct a shared global model $\mathbf{w}$ that has a generalization capability to predict well on any unseen target domain $\mathcal{T}$.

**Background.** Let $s \in \mathbb{R}^{C \times H \times W}$ be the feature of a sample which is obtained at a specific layer of the neural network. Here, $C$, $H$, $W$ denote the dimensions of channel, height, width, respectively. Given the feature $s$ of a specific data sample, the channel-wise mean $\mu(s) \in \mathbb{R}^C$ and the channel-wise standard deviation $\sigma(s) \in \mathbb{R}^C$ can be written as

$$\mu(s)_c = \frac{1}{HW} \sum_{h=1}^{H} \sum_{w=1}^{W} s_{c,h,w}, \quad \sigma^2(s)_c = \frac{1}{HW} \sum_{h=1}^{H} \sum_{w=1}^{W} (s_{c,h,w} - \mu(s)_c)^2, \tag{1}$$

respectively. These variables are known as *style statistics* as they contain style information of an image in CNNs [10]. Based on these style statistics, various style-augmentation schemes such as MixStyle [39] or DSU [21] have been proposed in the literature targeting a centralized setup.

**Overview of approach.** Fig. 1 provides an overview of the problem setup and our StableFDG algorithm. As in conventional FL, the training process consists of multiple global rounds, which we index $t = 1, 2, \ldots, T$. In the beginning of round $t$, a selected set of clients download the current global model $\mathbf{w}_t$ from the server. Before local training begins, each client $n$ computes its own style information $\Phi_n = [\mu_n, \sigma_n, \Sigma_n(\mu), \Sigma_n(\sigma)]$ using its local dataset according to (2), which will be clarified soon. This information is sent to the server, and the server shares these information with other clients to compensate for the lack of styles or domains in each client. During the local update process, each client selectively shifts the

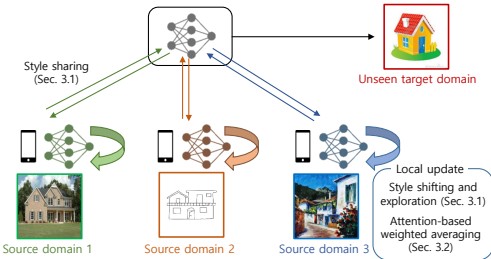

Figure 1: Overview of our problem setup and algorithm for federated domain generalization. Each client can have a single source domain as described in Fig. 1, or even multiple source domains in its local dataset.

styles of the original data in the mini-batch to the new style (received from the server) via adaptive instance normalization (AdaIN) [10], to improve domain diversity (inner box in Fig. 2b). After this style sharing and shifting process, each client performs style exploration via feature-level oversampling to further expose the model to novel styles beyond the current source domains of each client (outer box in Fig. 2b). Finally, at the output of the feature extractor, we apply our attention-based

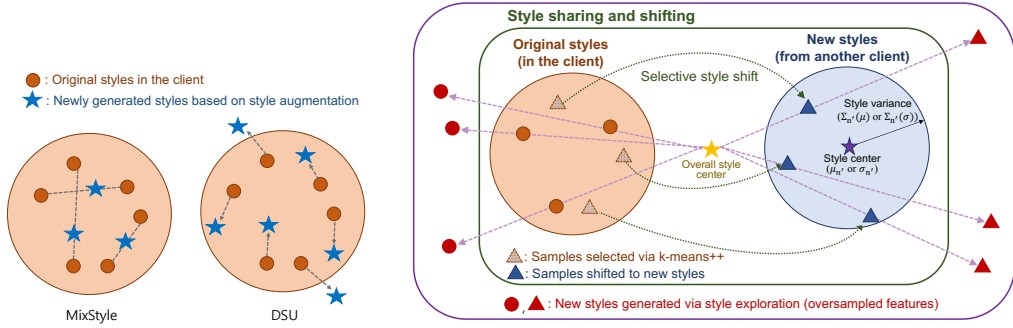

(a) Existing style-based DG baselines      (b) Proposed style-based learning for federated DG

Figure 2: **Proposed style-based learning strategy (Sec. 3.1):** Compared to existing style-based DG methods that rely on interpolation or style shift near the original style of each sample, our style-based learning (i) effectively utilizes other FL clients' styles based on style sharing and shifting, and (ii) enables the model to further explore a wider style space via feature-level oversampling and extrapolation, handling the issue of domain-limited FL settings.

feature highlighter to extract common/important feature information within each class and emphasize them for better generalization (Fig. 3). When local updates are finished, the server aggregates the client models and proceeds to the next round.

In the following, we first describe our style-based learning in Sec. 3.1, which determines how the style information is shared and utilized in each FL client, and how style exploration is conducted, to overcome the lack of styles in each client. In Sec. 3.2, we show how attention mechanism is utilized to capture the essential characteristics of each class for better generalization in data-poor FL setups.

## 3.1 Style-based Learning for Federated DG

The main components of our style-based learning are style-sharing, selective style-shifting, and style-exploration, each having its own role to handle the style limitation problem at each client. As illustrated in Fig. 2, this fundamental challenge of federated DG is not directly resolvable using existing style-augmentation DG methods that can only explore limited areas in the style-space.

**Step 1: Style information sharing.** Based on the data samples in its local dataset $\mathcal{S}_n$, at a specific layer of the model, each client $n$ computes the average of channel-wise means, $\mu_n$, and the variance of the channel-wise means, $\Sigma_n(\mu)$, as follows:

$$\mu_n = \frac{1}{\rho_n} \sum_{i=1}^{\rho_n} \mu(s_i^n), \quad \Sigma_n(\mu) = \frac{1}{\rho_n} \sum_{i=1}^{\rho_n} (\mu_n - \mu(s_i^n))^2, \tag{2}$$

where $s_i^n \in \mathbb{R}^{C \times H \times W}$ is the feature of the $i$-th sample in the $n$-th client's local dataset, and the square operation $(\cdot)^2$ works in an element-wise manner. Similarly, the average and the variance of channel-wise standard deviations are computed as $\sigma_n = \frac{1}{\rho_n} \sum_{i=1}^{\rho_n} \sigma(s_i^n)$, $\Sigma_n(\sigma) = \frac{1}{\rho_n} \sum_{i=1}^{\rho_n} (\sigma_n - \sigma(s_i^n))^2$, respectively. Here, $\mu_n$ and $\sigma_n$ represent the center of the style statistics of client $n$, while $\Sigma_n(\mu)$ and $\Sigma_n(\sigma)$ show how these style statistics of client $n$ are distributed around the center. Now we define

$$\Phi_n = [\mu_n, \sigma_n, \Sigma_n(\mu), \Sigma_n(\sigma)] \tag{3}$$

as the *style information* representing the domain identity of the $n$-th client in the style-space. Each client $n$ sends $\Phi_n$ to the server, and the server randomly shares the style information to other clients in a one-to-one manner; client $n$ receives $\Phi_{n'}$ that belongs to client $n'$ ($n \neq n'$) without overlapping with other clients. By compensating for the lack of styles in each FL client, this style-sharing process is the first step that provides an opportunity for each model to get exposed to new styles (blue region in Fig. 2b) beyond the client's original styles (orange region in Fig. 2b).

**Step 2: Selective style shifting.** Suppose client $n$ received $\Phi_{n'}$ from the server. Now the question is, how should each client utilize this additional style information during training to improve domain/style diversity? Our idea is to selectively shift the styles of the samples from the original style to the

new style to effectively balance between the original/new source domains. To this end, given a mini-batch with size $B$, each client runs $k$-means++ with $k = B/2$ in the style-space for one iteration and selects $B/2$ cluster centers. This enables each client to choose the $B/2$ styles that are similar to the remaining $B/2$ styles, so that the model can get exposed to new styles while not losing the performance on the original styles. The selected $B/2$ samples keep their original styles, while for the remaining $B/2$ samples, we shift the style of their feature $s$ to the new style via AdaIN [10] as $f(s) = (\sigma_{n'} + \epsilon_\sigma \Sigma_{n'}(\sigma)) \left( \frac{s - \mu(s)}{\sigma(s)} \right) + (\mu_{n'} + \epsilon_\mu \Sigma_{n'}(\mu))$, where $f(s)$ is the new feature shifted from $s$ and $\epsilon_\mu \sim \mathcal{N}(0, \Sigma_n(\mu)), \epsilon_\sigma \sim \mathcal{N}(0, \Sigma_n(\sigma))$ are the values sampled from normal distributions. Then, the mini-batch applied with new styles in $\Phi_{n'}$ is forwarded to the next layer. The inner box in Fig. 2b shows how style shifting is performed in client $n$ based on the new style information $\Phi_{n'}$.

Overall, based on the shared style statistics in Step 1, our style shifting in Step 2 balances between the original source domain and the new source domain via $k$-means++ for better generalization, which cannot be achieved by previous methods in Fig. 2a that rely on interpolation or style-shift near the original style. In the following, we describe our style exploration that can further resolve the style-limitation problem based on feature-level oversampling and extrapolation.

**Step 3: Feature-level oversampling.** Let $s^n \in \mathbb{R}^{B \times C \times H \times W}$ be a mini-batch of features in client $n$ at a specific layer, obtained after Steps 1 and 2 above. Here, we oversample the features by the mini-batch size $B$ in a class-balanced manner; the samples belonging to minority classes are compensated as balanced as possible up to size $B$. For example, suppose that the number of samples for classes $a$, $b$, $c$ are 3, 2, 1, respectively in the mini-batch. Using oversampling size of 6, we oversample by 1, 2, 3 data points for classes $a$, $b$, $c$, respectively, to balance the mini-batch in terms of classes. Based on this, we obtain a new oversampled mini-batch $\tilde{s}^n$ with size $B$, and concatenate it with the original mini-batch as follows: $\hat{s}^n = [s^n, \tilde{s}^n]$. This not only mitigates the class-imbalance issue in each client but also provides a good platform for better style exploration; the oversampled features are utilized to explore a wider style-space beyond the current source domains, as we will describe in Step 4. The oversampling size can be adjusted depending on the clients' computation/memory constraints.

**Step 4: Style exploration.** In order to further expose the model to a wider variety of styles, we transfer the styles of tensors in $\tilde{s}^n$ (i.e., the set of oversampled features) to novel styles beyond the style of each client. Let $\tilde{s}_i^n$ be the feature of $i$-th sample in the oversampled mini-batch $\tilde{s}^n$. We obtain the new styles by extrapolating the original styles in $\tilde{s}^n$ around the average of channel-wise mean $\mu_n(\hat{s}^n)$ and the average of standard deviations $\sigma_n(\hat{s}^n)$ computed on the concatenated mini-batch $\hat{s}^n = [s^n, \tilde{s}^n]$ as

$$\mu_{new}(\tilde{s}_i^n) = \mu(\tilde{s}_i^n) + \alpha \cdot (\mu(\tilde{s}_i^n) - \mu_n(s^n)), \tag{4}$$

$$\sigma_{new}(\tilde{s}_i^n) = \sigma(\tilde{s}_i^n) + \alpha \cdot (\sigma(\tilde{s}_i^n) - \sigma_n(s^n)), \tag{5}$$

where $\alpha$ is the *exploration level*. We perform AdaIN to shift the style of $\tilde{s}_i^n$ to the new style statistics $\mu_{new}(\tilde{s}_i^n)$ and $\sigma_{new}(\tilde{s}_i^n)$. If $\alpha = 0$, the styles remain unchanged, and as $\alpha$ increases, the styles are shifted farther from the center. The outer box in Fig. 2b describes the concept of our style exploration.

**Step 5: Style augmentation.** After style exploration, we can apply existing style-augmentation methods during training. In this work, we mix the style statistics of the entire samples in $\hat{s}^n$ to generate diverse domains as in [39] as $\mu_{new}(\hat{s}_i^n) = \lambda \cdot \mu(\hat{s}_i^n) + (1 - \lambda) \cdot \mu(\hat{s}_j^n)$ and $\sigma_{new}(\hat{s}_i^n) = \lambda \cdot \sigma(\hat{s}_i^n) + (1 - \lambda) \cdot \sigma(\hat{s}_j^n)$, where $\hat{s}_i^n$ and $\hat{s}_j^n$ are arbitrary two samples in $\hat{s}^n$ and $\lambda$ is a mixing parameter sampled from the beta distribution. Below, we wish to highlight two important points.

**Remark 1 (Privacy).** It is already well-known that there is an inherent clustering of samples based on their domains in the style-space, regardless of their labels [39]. This indicates that label information is not contained in the style statistics, resolving privacy issues. Note that some prior works on federated DG [25, 2] also adopt sharing style information between clients, but in different ways (see Sec. 2).

**Remark 2.** By enabling the model to explore a wider region in the style space based on the exploration level $\alpha$, our style exploration in Step 4 is especially beneficial when the target domain is significantly far from the source domains (e.g., Sketch domain in PACS dataset). Existing style-based DG methods (e.g., MixStyle, DSU) are ineffective in this case as they can explore only some limited areas near the original styles in each client (Fig. 2a vs. Fig. 2b), which leads to performance degradation especially in data-poor FL scenarios. It turns out that our scheme has significant performance improvements over these baselines even with a rather arbitrarily chosen $\alpha$, as we will see in Sec. 4.3.

## 3.2 Attention-based Feature Highlighter

In this subsection, we describe the second component of our solution, the attention-based feature highlighter, which operates at the output of the feature extractor to tackle the remaining challenge of federated DG: limited generalization in each client due to the lack of data. To handle this issue, we start from our key intuition that the images from the same class have inherently common characteristics regardless of domains to which they belong (e.g., Fig. 3). Based on this insight, we take advantage of attention to find the essential characteristics of images in the same class and emphasize them, which turns out to be very effective in data-poor FL scenarios.

**Attention score.** Consider the $i$-th data sample of client $n$. Given the feature $z_i^n \in \mathbb{R}^{C \times H \times W}$ obtained from the output of the feature extractor, we flatten it to a two-dimensional tensorop $X_i \in \mathbb{R}^{C \times HW}$ with a size of $(C, HW)$, where we omit the client index $n$ for notational simplicity. Now consider another $X_j$ ($j \neq i$) in a mini-batch that belongs to the same class as $X_i$, where the domains of $X_i$ and $X_j$ can be either same or different. Inspired by the attention mechanism [32], we measure the spatial similarity $S \in \mathbb{R}^{HW \times HW}$ between $X_i$ and $X_j$ as follows:

$$S := \text{Sim}(X_i, X_j) = (\theta_q X_j)^T (\theta_k X_i), \tag{6}$$

where $\theta_q \in \mathbb{R}^{d \times C}$, $\theta_k \in \mathbb{R}^{d \times C}$ are the learnable parameters trained to extract important information in each class from the given samples and $d$ is the embedding size of queries $Q = \theta_q X_j$ and keys $K = \theta_k X_i$. Then, we reshape $S \in \mathbb{R}^{HW \times HW}$ to $S_r \in \mathbb{R}^{HW \times H \times W}$; the $(m, n)$-th spatial feature ($\in \mathbb{R}^{HW \times 1 \times 1}$) of $S_r$ represents the similarity between the $(m, n)$-th spatial feature of the key feature $X_i$ and the overall spatial feature of the query feature $X_j$. Then by taking the mean of $S_r$ along the first dimension, we obtain the *attention score* $a_s$, which represents how the spatial feature at each location of key $X_i$ is similar to the overall features of query $X_j$: $a_s = \text{mean}(S_r) \in \mathbb{R}^{H \times W}$. A higher score $a_s$ indicates a higher similarity. Finally, we normalize the attention score through softmax function so that the total sum is one: $a_s \leftarrow \text{softmax}(a_s) \in \mathbb{R}^{H \times W}$.

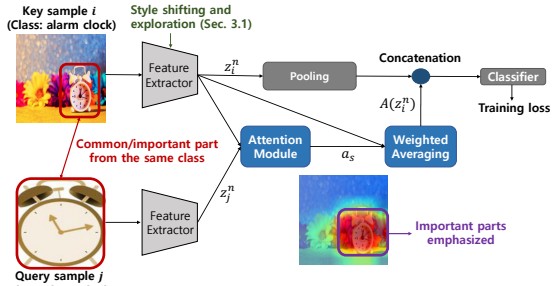

Figure 3: **Proposed attention-based feature highlighter (Sec. 3.2):** Our attention-based learning captures important characteristics within each class (regardless of the domain) for better generalization.

**Attention-based weighted averaging.** Based on the attention score, we take the weighted average of $z_i \in \mathbb{R}^{C \times H \times W}$ to generate an attention feature $A(z_i) \in \mathbb{R}^C$ in which the important parts of the *key image*, having common characteristics with the *query image* (in the same class), are emphasized:

$$A(z_i)_c = \sum_{h=1}^{H} \sum_{w=1}^{W} (a_s)_{h,w} (z_i)_{c,h,w}. \tag{7}$$

As shown in Fig. 3, this attention feature is concatenated with the last embedding feature (e.g., after the global average pooling in ResNet) before the classifier, and it goes through the classifier to compute the loss during training. One important thing to note is that $\theta_q$ and $\theta_k$ are trained so that the common features of query and key images become close in the embedding space while unimportant factors such as backgrounds are effectively distanced (i.e., less emphasized).

When implementing attention-based weighted averaging in practice, instead of directly adopting equation (6), we modify the similarity function using the query of its own as

$$\text{Sim}_{\text{mix}}(X_i, X_j) = \left( \frac{\theta_q X_j + \theta_q X_i}{2} \right)^T (\theta_k X_i), \tag{8}$$

to avoid performance degradation when there is little commonality between key and query images. In other words, StableFDG takes advantage of both cross-attention and self-attention, enabling the model to extract and learn important characteristics across images (via cross-attention), and within the image (via self-attention). A more detailed analysis on (8) can be found in Appendix.

**Inference.** During testing, given a new test sample, we compute the spatial similarity of the test sample itself as $\text{Sim}(X_i, X_i)$ based on self-attention, take the weighted average, and concatenate the features to make a prediction.

**Remark 3.** It is interesting to note that we can adopt *self-attention* during testing. This is because $\theta_q$ and $\theta_k$ are trained to capture the essential characteristics of each class, which enables the attention module to effectively find the important parts of individual samples via self-attention (e.g., see heat map visualization in Fig. 3). Moreover, with only 0.44% of additional model parameters for the attention module, this strategy turns out to be very effective in learning domain-invariant characteristics of classes in data-poor FL setups, handling the key challenge of federated DG.

### 3.3 StableFDG

Finally, we put together StableFDG. In each FL round, the clients first download the global model from the server and perform style sharing, shifting, and exploration according to 3.1, which are done in the early layers of CNNs where the style information is preserved. Then, at the output of the feature extractor, attention-based weighted averaging is applied according to Sec. 3.2. These two components have their own roles and work in a complementary fashion to handle the challenging DG problem in FL; our style-based strategy is effective in improving the domain diversity, while our attention-based method can directly capture the domain-invariant characteristics of each class. After the local update process, the server aggregates the client models and proceeds to the next round.

**Remark 4 (Computational complexity).** The computational complexity of StableFDG depends on the oversampling size in Sec. 3.1 and the attention module size in Sec. 3.2, which could be controlled depending on the resource constraints of clients. We show in Sec. 4 that StableFDG achieves the state-of-the-art performance with (i) minimal oversampling size and (ii) negligible cost of attention module. A more detailed discussion on the computational complexity is in Appendix.

## 4 Experimental Results

### 4.1 Experimental Setup

**Datasets.** We consider five datasets commonly adopted in DG literature: PACS [14], VLCS [6], Digits-DG [38], Office-Home [33], and DomainNet [30]. PACS consists of 7 classes from 4 different domains, while VLCS contains 4 domains with 5 classes. Digits-DG is composed of 4 different digit datasets, MNIST [13], MNIST-M [8], SVHN [27], SYN [8], each corresponding to a single domain. Office-Home consists of 65 classes from 4 domains, while DomainNet has 345 classes from 6 domains. The DomainNet results are reported in Appendix.

**Data partitioning for FL.** When evaluating the model performance, we follow the conventional leave-one-domain-out protocol where one domain is selected as a target and the remaining domains are utilized as sources. Compared to the centralized setup, in FL, the source domains are distributed across the clients. We consider a setup with $N = 30$ clients and distribute the training set into two different ways: *single-domain data distribution* and *multi-domain data distribution* scenarios. In a single-domain setup, we let each client to have training data that belong to a single source domain. Since there are three different source domains during training (except DomainNet), the training data of each domain is distributed across 10 clients uniformly at random. In a multi-domain distribution setup, each client can have multiple domains, but the domain distribution within each client is heterogeneous. For each domain, we sample the heterogeneous proportion from Dirichlet distribution with dimension $N = 30$ and parameter of 0.5, and distribute the train samples of each domain to individual clients according to the sampled proportion. In Appendix, we also report the results with $N = 3$ following the settings of prior works in federated DG [2, 28].

**Implementation.** Following [39], we utilize ResNet-18 pretrained on ImageNet as a backbone while the results on ResNet-50 are reported in Sec. 4.3. The exploration level $\alpha$ is set to 3 for all experiments regardless of datasets. For our attention module, we set the embedding size $d$ of queries $Q$ and keys $K$ to 30, where $Q$ and $K$ matrices are extracted from the output of the last residual block using $1 \times 1$ convolution such that the channel size of each output is 30. When the attention module is applied, the input dimension of the classifier becomes twice as large since we concatenate the attention feature with the last embedding feature before the classifier. The number of additional model parameters for the attention module is only 0.44% of the entire model. FL is performed for 50 global rounds and we trained the local model for 5 epochs with a mini-batch size of 32. Among a

| Methods | PACS | | | | | VLCS | | | | |
|---|---|---|---|---|---|---|---|---|---|---|
| | Art | Cartoon | Photo | Sketch | Avg. | Caltech | LabelMe | Pascal | Sun | Avg. |
| FedAvg [26] | 73.67 | 70.87 | 90.27 | 55.70 | 72.63 | 93.75 | 59.30 | 70.05 | 69.90 | 73.25 |
| FedBN [22] | 78.42 | 70.9 | 90.96 | 54.07 | 73.59 | 94.81 | 58.59 | 72.06 | 70.36 | 73.96 |
| MixStyle [39] | 79.10 | 76.30 | 90.10 | 60.63 | 76.53 | 95.20 | 60.40 | 72.10 | 69.93 | 74.41 |
| DSU [21] | 80.43 | 75.70 | 92.60 | 69.87 | 79.65 | 96.13 | 58.77 | 71.80 | 71.87 | 74.64 |
| CCST [2] | 71.35 | 72.40 | 88.65 | 64.10 | 74.13 | 92.50 | 61.20 | 68.20 | 66.50 | 72.10 |
| FedDG [25] | 71.20 | 71.40 | 90.70 | 59.20 | 73.13 | 95.3 | 57.5 | 72.8 | 69.8 | 73.85 |
| FedSR [28] | 76.40 | 71.25 | 93.25 | 60.55 | 75.36 | 92.10 | 60.50 | 70.75 | 71.65 | 73.75 |
| **StableFDG** (ours) | 84.10 | 78.57 | 95.40 | 72.73 | **82.70** | 98.13 | 59.20 | 73.60 | 70.27 | **75.30** |

(a) PACS and VLCS datasets.

| Methods | Office-Home | | | | | Digits-DG | | | | |
|---|---|---|---|---|---|---|---|---|---|---|
| | Art | Clipart | Product | Real | Avg. | MNIST | MNIST-M | SVHN | SYN | Avg. |
| FedAvg [26] | 57.27 | 48.23 | 72.77 | 74.60 | 63.22 | 98.05 | 70.95 | 68.95 | 86.40 | 81.09 |
| FedBN [22] | 57.56 | 48.13 | 72.65 | 74.57 | 63.23 | 97.33 | 72.68 | 71.77 | 85.36 | 81.79 |
| MixStyle [39] | 56.05 | 51.55 | 70.95 | 73.25 | 62.95 | 97.75 | 74.25 | 70.85 | 85.50 | 82.09 |
| DSU [21] | 58.55 | 52.60 | 71.60 | 73.15 | 63.98 | 98.10 | 75.60 | 70.47 | 85.80 | 82.49 |
| CCST [2] | 51.3 | 51.75 | 70.2 | 70.3 | 60.89 | 95.10 | 62.80 | 56.60 | 74.90 | 72.35 |
| FedDG [25] | 57.6 | 48.1 | 72.55 | 74.33 | 63.15 | 97.97 | 72.13 | 71.03 | 87.87 | 82.25 |
| FedSR [28] | 57.8 | 48.1 | 72.1 | 74.2 | 63.05 | 98.00 | 73.00 | 68.50 | 86.70 | 81.55 |
| **StableFDG** (ours) | 57.57 | 54.30 | 72.33 | 74.97 | **64.79** | 97.23 | 74.53 | 72.95 | 85.85 | **82.64** |

(b) Office-Home and Digits-DG datasets.

Table 1: **Main result 1 (single-domain data distribution):** Each client has one source domain in its local data. The proposed StableFDG achieves the best generalization, underscoring its effectiveness.

total of $N = 30$ clients, 10 clients participate in each global round. All reported results are averaged over three random seeds.

**Where to apply style-based modules.** Inspired by [39, 21], we apply our style-based modules with a probability of 0.5 at specific layers. In particular, style sharing and shifting are executed at the first residual block of ResNet with a probability 0.5, while the style exploration module is performed at the first or second or third residual blocks independently with probability 0.5 after style sharing/shifting.

**Baselines. (i) FL baselines:** First, we consider FedAvg [26] and FedBN [22], which are the basic FL baselines not specific to DG. **(ii) DG baselines applied to each FL client:** We apply MixStyle [39] during the local update process of each client and aggregate the model via FedAvg. Similarly, we also apply DSU [21] at each client and then perform FedAvg to compare with our work. **(iii) Federated DG baselines:** Among DG schemes tailored to FL, we implement the recently proposed CCST [2], FedDG [25], FedSR [28] and evaluate the performance. For a fair comparison, we reproduced all the baselines in accordance with our experimental setup. The *augmentation level* in CCST, a hyperparameter that controls the amount of augmented images, is set to 3 as in the original paper [2]. The hyperparameters in FedSR, that control the regularization losses, are tuned to achieve the best performance in our setup. Except for these, we adopted the parameter values in the original papers.

### 4.2 Main Experimental Results

**Single-domain data distribution.** Table 1 shows our results in a single-domain data distribution setup. We have the following observations. Compared to the previous results provided in the centralized DG works [39, 21], the performance of each method is generally lower. This is due to the limited numbers of styles and data samples in each FL client, which restricts the generalization performance of individual client models. It can be seen that most of the baselines perform better than FedAvg and FedBN that do not tackle the DG problem. The proposed StableFDG achieves the best average accuracy for all benchmark datasets, where the gain is especially large in PACS having large shifts between domains. In contrast to our scheme, the prior works [2, 25, 28] targeting federated DG show marginal performance gains relative to FedAvg in our practical experimental setup with (i) more clients (which results in less data in each client) and (ii) partial client participations.

**Multi-domain data distribution.** In Table 2, we report the results in a multi-domain data distribution scenario. Compared to the results in Table 1, most of the schemes achieve improved performance in Table 2. This is because each client has multiple source domains, and thus providing a better platform for each client model to gain generalization ability. The proposed StableFDG still performs the best, demonstrating the effectiveness of our style and attention based learning strategy for federated DG.

| Methods | | Office-Home | | | | VLCS | | | | |
|---|---|---|---|---|---|---|---|---|---|---|
| | | Art | Clipart | Product | Real | Avg. | Caltech | LabelMe | Pascal | Sun | Avg. |
| FedAvg [26] | | 57.70 | 48.30 | 72.87 | 75.33 | 63.55 | 93.65 | 61.10 | 72.55 | 65.40 | 73.18 |
| FedBN [22] | | 57.07 | 48.32 | 72.31 | 74.57 | 63.07 | 94.34 | 62.61 | 69.89 | 69.04 | 73.97 |
| MixStyle [39] | | 55.87 | 52.03 | 71.10 | 74.20 | 63.30 | 95.20 | 60.77 | 73.90 | 66.73 | 74.15 |
| DSU [21] | | 58.60 | 52.80 | 71.63 | 74.00 | 64.26 | 96.87 | 60.23 | 72.97 | 68.97 | 74.76 |
| CCST [2] | | 52.2 | 52.2 | 70.6 | 72.3 | 61.83 | 96.70 | 60.40 | 71.40 | 65.00 | 73.38 |
| FedDG [25] | | 57.9 | 48.6 | 73.2 | 75.0 | 63.68 | 96.2 | 60.7 | 72.4 | 67.3 | 74.15 |
| FedSR [28] | | 58.1 | 48.2 | 72.5 | 75.4 | 63.55 | 92.60 | 60.80 | 72.15 | 68.30 | 73.46 |
| **StableFDG** (ours) | | 57.87 | 54.20 | 73.10 | 75.00 | **65.04** | 98.50 | 60.07 | 74.40 | 69.43 | **75.60** |

(a) Office-Home and VLCS datasets.

| Methods | | Art | Cartoon | Photo | Sketch | Avg. |
|---|---|---|---|---|---|---|
| FedAvg [26] | | 74.87 | 74.53 | 95.30 | 63.37 | 77.02 |
| FedBN [22] | | 77.00 | 76.83 | 95.45 | 67.01 | 79.07 |
| MixStyle [39] | | 80.07 | 77.53 | 96.23 | 67.40 | 80.31 |
| DSU [21] | | 80.53 | 76.30 | 95.37 | 70.93 | 80.78 |
| CCST [2] | | 75.53 | 75.80 | 93.53 | 71.13 | 79.00 |
| FedDG [25] | | 75.35 | 75.85 | 95.65 | 61.05 | 76.98 |
| FedSR [28] | | 73.83 | 74.83 | 95.53 | 66.03 | 77.56 |
| **StableFDG** (ours) | | 83.97 | 79.10 | 96.27 | 75.67 | **83.75** |

(b) PACS dataset.

Table 2: **Main result 2 (multi-domain data distribution):** Each client has multiple source domains in its local dataset. The results are consistent with the single-domain scenario in Table 1.

## 4.3 Ablation Studies and Discussions

**Effect of each component.** To see the effect of each component of StableFDG, in Table 3, we apply our style-based learning and attention-based learning one-by-one in a multi-domain data distribution setup using PACS. We compare our results with style-augmentation DG baselines, MixStyle [39] and DSU [21]. By applying only our style-based learning, StableFDG already

| Methods | Art | Cartoon | Photo | Sketch | Avg. |
|---|---|---|---|---|---|
| MixStyle [39] | 80.07 | 77.53 | 96.23 | 67.40 | 80.31 |
| DSU [21] | 80.53 | 76.30 | 95.37 | 70.93 | 80.78 |
| StableFDG (only style) | 82.62 | 79.01 | 95.57 | 74.47 | 82.92 |
| StableFDG (only attention) | 79.98 | 78.58 | 95.75 | 71.35 | 81.41 |
| StableFDG (both) | 83.97 | 79.10 | 96.27 | 75.67 | **83.75** |

Table 3: Effect of each component of StableFDG.

outperforms prior style-augmentation methods. Furthermore, by adopting only one of the proposed components, our scheme performs better than all the baselines in Table 2. Additional ablation studies using other datasets are reported in Appendix.

**Effect of hyperparameters.** In DG setups, it is generally impractical to tune the hyperparameter using the target domain, because there is no information on the target domain during training. Hence, we used a fixed exploration level $\alpha = 3$ throughout all experiments without tuning. In Fig. 4, we observe how the hyperparameters affect the target domain performance on PACS. In the first plot of Fig. 4, if $\alpha$ is too small, the performance is relatively low since

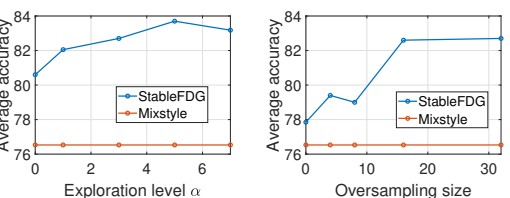

Figure 4: Effects of exploration level $\alpha$ (left) and oversampling size (right) in StableFDG.

the model is not able to explore novel styles beyond the client's source domains. If $\alpha$ is too large, the performance could be slightly degraded because the model would explore too many redundant styles. The overall results show that StableFDG still performs better than the baselines with an arbitrarily chosen $\alpha$, which is a significant advantage of our scheme in the DG setup where hyperparameter tuning is challenging. The second plot of Fig. 4 shows how the oversampling size (introduced in Step 3 of Sec. 3.1) affects the DG performance. StableFDG still outperforms the baseline with minimal oversampling size, indicating that other components of our solution (style sharing/shifting and attention-based components) are already strong enough. The size of oversampling can be determined depending on the clients' computation/memory constraints, with the cost of improved generalization.

**Performance in a centralized setup.** Although our scheme is tailored to federated DG, the ideas of StableFDG can be also utilized in a centralized setup. In Table 4a, we study the effects of our style and attention based strategies in a centralized DG setting using PACS, while the other settings are

| Methods | Art | Cartoon | Photo | Sketch | Avg. |
|---|---|---|---|---|---|
| MixStyle [39] | 81.61 | 78.83 | 96.77 | 72.29 | 82.38 |
| DSU [21] | 78.84 | 79.56 | 95.21 | 79.39 | 83.25 |
| **StableFDG** (ours) | 85.02 | 79.65 | 96.38 | 78.35 | **84.85** |

(a) Performance in a centralized DG setup.

| Methods | Art | Cartoon | Photo | Sketch | Avg. |
|---|---|---|---|---|---|
| MixStyle [39] | 87.51 | 81.12 | 97.48 | 68.39 | 83.63 |
| DSU [21] | 86.48 | 81.22 | 97.21 | 73.99 | 84.73 |
| **StableFDG** (ours) | 90.01 | 83.29 | 98.02 | 79.47 | **87.70** |

(b) Performance using ResNet-50.

Table 4: The applicability of StableFDG in a centralized DG setup (Table 4a) and performance using a larger model (Table 4b) on the PACS dataset.

the same as in the FL setup. The results demonstrate that the proposed ideas are not only specific to data-poor FL scenarios but also have potentials to be utilized in centralized DG settings.

**Performance with ResNet-50.** In Table 4b, we also conduct experiments using ResNet-50 on PACS dataset in the multi-domain data distribution scenario. Other settings are exactly the same as in Table 2. The results further confirm the advantage of StableFDG with larger models.

**Attention score visualization.** To gain an intuitive understanding of the effect of our attention-based learning, in Fig. 5, we visualize the score maps obtained via our attention module at testing. The score maps are interpolated so that it has the same size as the original image. A warmer color indicates a higher value. It can be seen that our attention module highlights important parts of each class even in the presence of unrelated backgrounds.

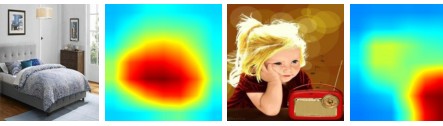

Figure 5: Visualization of attention score maps of each input image (left: bed, right: radio).

**Additional results.** Other implementation details, comprehensive ablation studies for each component, discussions on complexity, and results on DomainNet dataset are in Appendix.

## 5   Conclusion

Despite the practical significance, the field of federated domain generalization is still in the early stage of research. In this paper, we proposed StableFDG, a new training strategy tailored to this unexplored area. Our style-based strategy enables the model to get exposed to various novel styles beyond each client's source domains, while our attention-based method captures and emphasizes the important/common characteristics of each class. Extensive experimental results confirmed the advantage of our StableFDG for federated domain generalization with data-poor FL clients.

**Limitations and future works.** StableFDG requires 0.45 % more communication load compared to FedAvg for sharing the attention module and style statistics, which is the cost for a better DG performance. Further developing our idea to tailor to centralized DG and extending our attention strategy to segmentation/detection DG tasks are also interesting directions for future research.

## Acknowledgments

This work was supported by the National Research Foundation of Korea (NRF) grant funded by the Korea government (MSIT) (No. NRF-2019R1I1A2A02061135), by IITP funds from MSIT of Korea (No. 2020-0-00626), by the National Science Foundation (NSF) under grants CNS-2146171 and CPS-2313109, and by the Defense Advanced Research Projects Agency (DARPA) under grant D22AP00168-00.

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

## A   Results on DomainNet Dataset

To demonstrate the effectiveness of our StableFDG on a larger dataset, we performed experiments on DomainNet dataset in a single-domain data distribution scenario. To this end, we utilize ResNet-50 pretrained on ImageNet. The number of global rounds and mini-batch size are set to 60 and 64, respectively. The remaining settings are the same as those in our main paper. The results in Table 5 show that our StableFDG consistently outperforms not only the centralized DG works but also the prior works on federated DG even with a more complex dataset.

| Methods | Clipart | Inforgraph | Painting | Quickdraw | Real | Sketch | Avg. |
|---|---|---|---|---|---|---|---|
| FedAvg [26] | 61.52 | 24.75 | 50.83 | 12.08 | 60.00 | 49.92 | 43.18 |
| FedBN [22] | 60.23 | 24.40 | 50.43 | 11.99 | 59.57 | 49.66 | 42.71 |
| MixStyle [39] | 61.39 | 24.33 | 51.41 | 13.07 | 57.90 | 51.40 | 43.25 |
| DSU [21] | 62.43 | 24.30 | 51.87 | 13.65 | 58.75 | 52.40 | 43.90 |
| FedDG [25] | 62.49 | 23.71 | 48.42 | 12.45 | 61.44 | 49.11 | 42.94 |
| FedSR [28] | 61.91 | 25.37 | 50.54 | 11.59 | 62.03 | 50.13 | 43.60 |
| **StableFDG** | 62.58 | 24.12 | 52.23 | 14.87 | 60.60 | 52.50 | **44.48** |

Table 5: Results on DomainNet dataset in a single-domain data distribution scenario.

## B   Discussion on Computational and Communication Costs

Table 6 compares the communication, computation, and average accuracy of different schemes on PACS in a multi-domain data distribution scenario. ResNet-18 is adopted as in our main manuscript. We first compare the uplink communication load of each client in a specific global round. Compared to FedAvg that only transmits the model in each round, our scheme requires additional communication burden for transmitting the style statistics and the attention module, which are negligible. We also compare the computation time by measuring the time required for local update at each client using an GTX 1080 Ti GPU. CCST [2] and FedDG [25] require large computation due to the increased amounts of data samples or multiple backpropagations for meta training. Our scheme requires additional computation caused by style exploration, attention module update, etc., which are the costs for better generalization to the unseen domain.

| Methods | Communication load | Computation time | Achievable average accuracy |
|---|---|---|---|
| FedAvg [26] | 44.98 MB | 4.57 sec | 77.02 % |
| CCST [2] | 44.98 MB | 9.03 sec | 79.00 % |
| FedDG [25] | 44.98 MB | 22.44 sec | 76.98 % |
| FedSR [28] | 44.98 MB | 4.59 sec | 77.56 % |
| StableFDG | 45.00 MB | 7.39 sec | **83.75** % |

Table 6: Computation and communication cost comparison.

## C   Experiments with Three Clients

Different from the prior works [2, 28] for federated DG adopting the usual setting where the number of clients equals the number of source domains, in our main paper, we introduce a more practical experimental setting for federated DG where the source data is distributed to more clients than the number of source domains. For a comparison with them in the same setting, we also provide additional experimental results in the setup with *number of clients = number of source domains*. Table 7 shows the results on PACS and Office-Home datasets with three clients in a single-domain data distribution scenario. It is confirmed from the results that our StableFDG also achieves better performance compared to the existing works [2, 28] in this simple setting.

| Methods | Art | Cartoon | Photo | Sketch | Avg. |
|---------|-----|---------|-------|--------|------|
| CCST [2] | 81.25 | 73.34 | 95.21 | 80.27 | 82.52 |
| FedSR [28] | 83.20 | 76.00 | 93.80 | 81.90 | 83.70 |
| **StableFDG** | 83.01 | 79.31 | 94.85 | 79.76 | **84.23** |

(a) Results on PACS dataset.

| Methods | Art | Clipart | Product | Real | Avg. |
|---------|-----|---------|---------|------|------|
| CCST [2] | 59.05 | 50.06 | 72.97 | 71.67 | 63.56 |
| FedSR [28] | 57.93 | 50.45 | 73.33 | 75.51 | 64.31 |
| **StableFDG** | 57.19 | 57.94 | 72.76 | 72.16 | **65.01** |

(b) Results on Office-Home dataset.

Table 7: Results in a single-domain data distribution scenario with three clients.

# D    Effect of Each Component of StableFDG

As mentioned in the main manuscript, we provide further ablation studies on the effect of each component in StableFDG using VLCS dataset. Table 8 shows that each component individually brings performance gain compared to FedAvg. Using both strategies achieves greater performance gains, confirming that the two proposed schemes work in a complementary fashion.

| Methods | Caltech | Labelme | Pascal | Sun | Avg. |
|---------|---------|---------|--------|-----|------|
| FedAvg [26] | $93.65 \pm 1.59$ | $61.10 \pm 2.08$ | $72.55 \pm 0.90$ | $65.40 \pm 0.28$ | $73.18 \pm 0.27$ |
| StableFDG (only style) | $98.19 \pm 0.81$ | $58.93 \pm 1.16$ | $75.19 \pm 0.67$ | $68.69 \pm 0.73$ | $75.25 \pm 0.20$ |
| StableFDG (only attetnion) | $94.10 \pm 1.22$ | $62.02 \pm 1.55$ | $72.26 \pm 0.78$ | $66.16 \pm 2.58$ | $73.64 \pm 0.89$ |
| StableFDG (both) | $98.50 \pm 0.17$ | $60.07 \pm 0.79$ | $74.40 \pm 1.88$ | $69.43 \pm 1.11$ | $\mathbf{75.61} \pm 0.71$ |

Table 8: Effect of each component on VLCS dataset in a multi-domain data distribution scenario. The reported results indicate (mean $\pm$ 95% confidence interval) over 3 random trials.

# E    Ablation Studies on Style-Based Learning

## E.1    Randomness in style sharing

In our style based learning, style sharing among clients is performed at random. However, one can think of the strategy where client $n$ receives the style information $\Phi_{n'}$ that has the largest distance with its own style information $\Phi_{n'}$ in the style space. Table 9 shows the corresponding result using PACS dataset in a multi-domain data distribution setup. Interestingly, it can be seen that the random selection adopted in this paper performs better, since most of the users generally tend to receive the same style statistics when using the *largest distance* strategy.

| Methods | Art | Cartoon | Photo | Sketch | Avg. |
|---------|-----|---------|-------|--------|------|
| Random sharing (current manuscript) | $83.97 \pm 1.25$ | $79.10 \pm 0.45$ | $96.27 \pm 0.36$ | $75.67 \pm 0.58$ | $83.75 \pm 0.23$ |
| Large distance | $83.01 \pm 1.43$ | $78.33 \pm 0.51$ | $96.19 \pm 0.76$ | $74.63 \pm 0.65$ | $83.04 \pm 0.30$ |

Table 9: Random sharing vs receiving the style that has the largest distance in style space. The reported results indicate (mean $\pm$ 95% confidence interval) over 3 random trials.

## E.2    Effect of the number of shared styles

We also compare the effect of number of styles received at each client in Table 10 using PACS dataset in a multi-domain data distribution setup. The performance increases as the number of received styles increases, with small additional communication load (the vector length of the style information is 128, which is negligible compared to the number of model parameters, which is 11,180,103).

| Methods | Art | Cartoon | Photo | Sketch | Avg. |
|---|---|---|---|---|---|
| No style sharing | $83.16 \pm 0.18$ | $78.54 \pm 1.11$ | $95.56 \pm 0.92$ | $74.64 \pm 1.06$ | $82.97 \pm 0.36$ |
| Receive 1 style (current manuscript) | $83.97 \pm 1.25$ | $79.10 \pm 0.45$ | $96.27 \pm 0.36$ | $75.67 \pm 0.58$ | $83.75 \pm 0.23$ |
| Receive 3 styles | $84.55 \pm 1.20$ | $78.67 \pm 0.59$ | $95.75 \pm 0.58$ | $76.77 \pm 0.27$ | $83.94 \pm 0.07$ |

Table 10: Effect of the number of shared styles. The reported results indicate (mean $\pm 95\%$ confidence interval) over 3 random trials.

### E.3 Effect of $k$-means++ in style shifting

In Section 3.1, we utilized the $k$-means++ as a tool for facilitating our key idea in style shifting, which is to effectively balance between the original source domain and the new source domain for better generalization; k-means++ plays a role to select the $B/2$ styles that are similar to the remaining $B/2$ styles in the mini-batch. By doing so, the model can explore new styles while not losing the performance on the original styles. To see this effect, we compare $k$-means++ vs. random sampling when selecting B/2 samples to be shifted, in Table 11. The results show that strategically selecting the B/2 samples to be shifted achieves better performance especially in the Sketch domain (1.89% gain) that has a large style gap with other domains. We believe that these results motivate and justify our design choice.

| Methods | Art | Cartoon | Photo | Sketch | Avg. |
|---|---|---|---|---|---|
| Shifting $B/2$ random styles | 83.69 | 79.61 | 95.99 | 73.78 | 83.27 |
| Shifting $B/2$ styles via $k$-means++ | 83.97 | 79.10 | 96.27 | 75.67 | **83.75** |

Table 11: Effect of $k$-means++ in style shifting on PACS dataset.

### E.4 Effect of class-balanced oversampling

In our main paper, we performed the class-balanced oversampling in the feature space to alleviate the class-imbalance issue during style exploration. To confirm the effectiveness of the class-balanced oversampling, we compare it with random oversampling under the same condition where only the style exploration is applied without other components. Table 12 shows the results on Office-Home dataset, where the class distribution is highly imbalanced in a multi-domain data distribution scenario. It can be seen that our class-balanced oversampling achieves higher performance over the simple random sampling, which validates the efficacy of mitigating the class imbalance problem in FL clients.

| Methods | Art | Cartoon | Photo | Sketch | Avg. |
|---|---|---|---|---|---|
| Random oversampling | 56.21 | 54.43 | 69.37 | 72.29 | 63.08 |
| Class-balanced oversampling | 57.20 | 53.15 | 71.2 | 74.23 | **64.10** |

Table 12: Effect of the class-balanced oversampling on Office-Home dataset in a multi-domain data distribution scenario.

### E.5 Effect of the operation probability in style-based learning

We conduct additional ablation studies on the probability value (defined as $p$ here) utilized to control the operation of the style sharing/shifting and style exploration modules. Larger $p$ means that our scheme is more likely to be activated. In Table 13, we provide results on various probability values in a single-domain data distribution scenario using PACS dataset. From the results, it is confirm that for all $p$ values, the proposed StableFDG outperforms existing baselines, demonstrating that our scheme can work well with an arbitrarily chosen probability $p$. In detail, when $p$ ranges from 0.3 to 0.7, the high performance is maintained while the performance decreases at both extreme probabilities ($p = 0.1$ and 0.9). Therefore, it is recommended for practitioners to select the $p$ in an appropriate range, avoiding extreme cases.

| Methods | Art | Cartoon | Photo | Sketch | Avg. |
|---|---|---|---|---|---|
| MixStyle [39] | 79.10 | 76.30 | 90.10 | 60.63 | 76.53 |
| DSU [21] | 80.43 | 75.70 | 92.60 | 69.87 | 79.65 |
| **StableFDG** ($p = 0.1$) | 82.70 | 78.30 | 95.30 | 75.30 | 82.90 |
| **StableFDG** ($p = 0.3$) | 84.50 | 79.50 | 96.00 | 75.70 | 83.93 |
| **StableFDG** ($p = 0.5$) | 83.10 | 79.50 | 96.40 | 76.00 | 83.75 |
| **StableFDG** ($p = 0.7$) | 82.40 | 78.10 | 95.70 | 76.30 | 83.13 |
| **StableFDG** ($p = 0.9$) | 81.40 | 78.80 | 95.90 | 73.60 | 82.70 |

Table 13: Effect of the probability value in our style-based learning on PACS in a single-domain data distribution scenario.

## E.6   Where to apply the style module

For implementation, style-based learning is applied only in the 1st, 2nd, 3rd blocks among 4 residual blocks in ResNet-18. Note that at the output of the 4th block, label information is dominant rather than style information, which results in degraded performance when style-based schemes are applied. This is confirmed by our new experiments in the table below. It can be seen from the results that if we consider the 4th residual block to apply our style-based learning, the performance gets degraded. This result confirms the intuition that style-based learning should be conducted at the earlier layers where style information is preserved.

| Methods | Art | Cartoon | Photo | Sketch | Avg. |
|---|---|---|---|---|---|
| Style exploration at 1st, 2nd, 3rd layers (main manuscript) | 84.10 | 78.57 | 95.40 | 72.73 | 82.70 |
| Style exploration at 1st, 2nd, 3rd, 4th layers | 82.99 | 78.54 | 94.13 | 73.35 | 82.25 |

Table 14: Ablation experiments on applying style-based learning at different layers (PACS dataset).

## F   Ablation Studies on Attention-based Feature Highlighter

### F.1   Effect of adopting both cross-attention and self-attention

In our main paper, the similarity metric in equation (6) adopts cross-attention, while the metric in equation (8) combines cross-attention and self-attention. When applying only the cross-attention-based metric in equation (6), we found that the similarity value could become low even when the two samples belong to the same class, in special cases. We handled this issue by adding the self-attention component as in equation (8). Intuitively, by doing this, the attention module is learning to extract the important features across images (via cross-attention), and within the image (via self-attention). Table 15 compares the performance of our StableFDG when using (i) self-attention alone, (ii) cross-attention alone (equation (6)), and (iii) both self and cross attentions at the same time (equation (8)), confirming the advantage of using self-attention and cross-attention together.

| Methods | Clipart | Infograph | Painting | Quickdraw | Real | Sketch | Avg. |
|---|---|---|---|---|---|---|---|
| StableFDG (with self-attention alone) | 61.77 | 24.88 | 48.28 | 14.15 | 59.78 | 52.41 | 43.55 |
| StableFDG (with cross-attention alone) | 61.24 | 24.97 | 50.55 | 14.67 | 61.12 | 50.44 | 43.83 |
| StableFDG (with self + cross) | 62.58 | 24.12 | 52.23 | 14.87 | 60.60 | 52.50 | **44.48** |

Table 15: Effects of similarity metrics using DomainNet dataset in a multi-domain data distribution scenario.

### F.2   Comparison using the same model size

Our attention module requires 0.44% of additional model parameters to perform the attention-based learning. For a fair comparison to see the effect of our attention-based learning, we consider a

different baseline with the same model size but without attention-based learning. Specifically, the baseline computes the attention score map using only additional convolutional operations and take the weighted average of the feature $z_i$ based on the attention score map. Table 16 shows the results using PACS dataset in a single-domain data distribution scenario. The results demonstrate that our attention-based learning achieves performance improvements on all four domains while playing a key role in capturing essential parts of the features.

| Methods | Art | Cartoon | Photo | Sketch | Avg. |
|---|---|---|---|---|---|
| Baseline (same model size) | 78.88 | 69.03 | 91.74 | 60.77 | 75.11 |
| Attention-based learning | 79.54 | 72.48 | 92.51 | 64.71 | **77.31** |

Table 16: Ablation study on attention-based learning using PACS dataset in a single-domain data distribution scenario.

### F.3  Effect of attention in a centralized setup

Now we provide answer to the following question: Instead of the FL setup we focused on, can attention provide benefits in the centralized DG setup? Table 17 shows the results with/without attention module in a centralized setup using PACS dataset. The results show that attention still provides performance improvements in the centralized setup by learning domain-invariant features, although the gain is slightly lower than the gain in the FL setup as shown in Table 3 of the main manuscript. These results indicate that the proposed attention-based learning indeed captures the domain-invariant characteristics of samples, while the scheme provides more benefits in the FL setup where each client is prone to overfitting due to lack of data.

| Methods | Art | Cartoon | Photo | Sketch | Avg. |
|---|---|---|---|---|---|
| StableFDG (centralized setup, without attention) | 84.15 | 79.45 | 96.21 | 77.09 | 84.23 |
| StableFDG (centralized setup, with attention) | 85.02 | 79.65 | 96.38 | 78.45 | 84.88 |

Table 17: Effect of proposed attention-based feature highlighter in a centralized DG setup using PACS dataset.

## G  Other Implementation Details

Our code is built upon the official code of [39] and [1]. During the local update process, we use SGD as an optimizer with a momentum of 0.9 and a weight decay of $5e^{-4}$. For PACS, Office-Home and VLCS, the learning rate is set to 0.001 and the cosine annealing is used as a scheduler. For Digits-DG, we set the learning rate to 0.02 and the learning rate is decayed by 0.1 every 20 steps. For our attention-based feature highlighter, at least two samples are required to be in the mini-batch of every client. When this condition is not met, additional samples are extracted from the corresponding client's local dataset to facilitate cross-attention.

**More detailed description on oversampling:** Let $s^n \in \mathbb{R}^{B \times C \times H \times W}$ be a mini-batch of features in client $n$ at a specific layer, obtained after Steps 1 and 2 in the main manuscript. Now given a fixed oversampling size, we oversample the features in the mini-batch to obtain $\tilde{s}^n$, so that the concatenated mini-batch $\hat{s}^n = [s^n, \tilde{s}^n]$ becomes class-balanced as much as possible. Consider a toy example where the number of samples for classes $a$, $b$, $c$ are 3, 2, 1, respectively in the mini-batch $s^n$. In this example, if the oversampling size is 3, we randomly choose one data point from class $b$ and two data points from class $c$ (in this case, the same data point is selected for two times with duplication) to obtain $\tilde{s}^n$, so that the concatenated mini-batch $\hat{s}^n = [s^n, \tilde{s}^n]$ becomes class-balanced. If the oversampling size is 1, we oversample one data point in class $c$ to make the concatenated mini-batch to be balanced as much as possible. If the oversampling size is 6, we oversample 1, 2, 3 samples from classes $a$, $b$, $c$, respectively to construct $\tilde{s}^n$. The concatenated mini-batch $\hat{s}^n = [s^n, \tilde{s}^n]$ is utilized for style-based learning and updating the model. This process not only mitigates the class-imbalance issue in each client but also provides a good platform for style exploration by oversampling the features. In our

work, we reported the results with oversampling size of $B$ (which is equal to the mini-batch size), while the effect of the oversampling size is also reported in Fig. 4 of the main manuscript: A larger oversampling size leads to a better performance, and more importantly, our StableFDG outperforms the baseline even without any oversampling.

## H    Visualization of Attention Score Maps

Finally in Fig. 6, we visualize the attention score maps of different input images during testing. The results confirm the effectiveness of our attention-based feature highlighter to focus on the important parts of each image from the unseen domain.

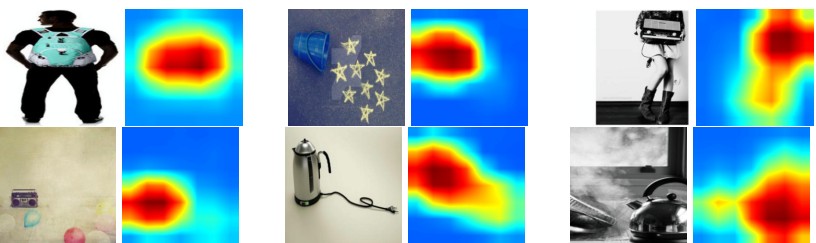

Figure 6: Visualization of attention score maps of each input image.

## I    Pseudo Code Algorithm of Stable FDG

Algorithm 1 summarizes the overall process of our StableFDG.

---

**Algorithm 1** StableFDG

---

**Input:** Initialized model $\mathbf{w}_0$, **Output:** Global model $\mathbf{w}_T$

 1: **for** each global round $t = 0, 1, ..., T-1$ **do**
 2:
 3:     **Stage 1.** Model download and style sharing
 4:     The server samples a set of participating clients $M_t$ and sends the global model $\mathbf{w}_t$ to the clients in $M_t$
 5:     **for** each device $n \in M_t$ **do**
 6:         Compute style information $\Phi_n$ according to the Step 1 in Sec. 3.1
 7:         Transmit $\Phi_n$ to the server
 8:     **end for**
 9:     The server shares $\{\Phi_n\}_{n \in M}$ to the clients in $M_t$ according to the Step 1 in Sec. 3.1
10:
11:     **Stage 2.** Local updates and model aggregation
12:     **for** each device $n \in M_t$ **do**
13:         **for** local epoch $= 1, 2..., E$ **do**
14:             (i) Style shifting according to the Step 2 in Sec. 3.1,
15:             (ii) Style exploration according to the Step 3 and 4 in Sec. 3.1
16:             (iii) Attention based weighted averaging according to Sec. 3.2
17:             Loss computation and model update
18:         **end for**
19:         The server updates the global model by aggregating the client models
20:     **end for**
21:
22: **end for**

---

