# OpenReview forum: "StableFDG: Style and Attention Based Learning for Federated Domain Generalization"
_NeurIPS.cc/2023/Conference — NeurIPS 2023 poster_

### Official Review · Reviewer_hWZZ · 2023-06-23

**Soundness:** 4 excellent
**Presentation:** 3 good
**Contribution:** 3 good
**Rating:** 6
**Confidence:** 3

**Summary:**

This paper proposes a style and attention based learning strategy for federated domain generalization which consists of two stages. The first stage shares style information among clients and then explorates and augments styles, and the second stage highlights the essential characteristics of images with attention mechanism. Experiments  show that the proposed method achieves a new state of the art.

**Strengths:**

- The style-based learning strategy is novel and intuitive.
- The paper is well written and easy to follow
- The proposed method is technically sound with superior results on the 5 datasets.

**Weaknesses:**

The detail of oversampling is not fully covered in the paper.

**Questions:**

- Why can eq.7. avoid performance degradation when there is little commonality between key and query images ?
- How is oversampling implemented. What if the dataset itself is not class balanced ?

**Limitations:**

The paper discusses its limitation in sec. 5.

---

> ### Author Rebuttal · Authors · 2023-08-08
>
> We thank the reviewer for the positive comments and  feedback. In particular, we  appreciate the reviewer’s acknowledgment on the novelty of our work.  Below, we reply to the comments raised by the reviewer.
>
> ### **[Question 1] Insights into equation (7)**
>
> We appreciate this comment. The similarity metric in equation 5 adopts cross-attention, while  the metric in equation 7  combines cross-attention and self-attention.  When applying only the cross-attention-based metric in equation 5, we found that  the similarity value could become low even when the two samples belong to the same class,   in special cases. We handled this issue by adding the self-attention component as in equation 7. Intuitively, by doing this, the attention module is   learning to extract the important features across images (via cross-attention), and within the image (via self-attention). The table below compares the performance of our StableFDG  when using equations 5 and 7 for similarity computations, confirming the advantage of using  self-attention and cross-attention together. In accordance with the reviewer's comment, we will make this clearer in the revised  manuscript.
>
> * Effect of similarity metrics:
> |Methods             | Art | Cartoon | Photo | Sketch | Avg |
> | :---------------- | :------: | :------: |:------: |:------: |:------: |
> | StableFDG (similarity metric in equation 5)       |   83.09 |  78.60 |  95.87 |  74.71 |  83.07
> | StableFDG (similarity metric in equation 7)        |   **83.97** |   **79.10**  |  **96.27** |  **75.67** |  **83.75**
>
>
> ### **[Question 2] How oversampling is implemented & the case with imbalanced dataset**
>
> **How feature-level oversampling is implemented:** Let $s^n\in \mathbb{R}^{B \times C \times H \times W}$ be a mini-batch of features in client $n$ at a specific layer,  obtained after   Steps 1 and 2 in the main manuscript.     Now given a fixed oversampling size, we oversample the features in the mini-batch to obtain  $\tilde{s}^n$, so that the concatenated mini-batch $\hat{s}^n =  [s^n, \tilde{s}^n ]$ becomes class-balanced as much as possible. Consider a toy example where the number of samples for classes  $a$, $b$, $c$ are 3, 2, 1, respectively in the mini-batch $s^n$. In this example, if the oversampling size is 3, we randomly choose   one data point from class $b$
>  and two data points from class  $c$ (in this case, the same data point is selected for two times with duplication) to obtain $ \tilde{s}^n$, so that the concatenated mini-batch $\hat{s}^n =  [s^n, \tilde{s}^n ]$ becomes class-balanced. If the oversampling size is 1, we oversample one  data point in class $c$ to make the concatenated mini-batch to be balanced as much as possible. If the oversampling size is 6, we oversample 1, 2, 3 samples from classes  $a$, $b$, $c$, respectively to construct $\tilde{s}^n$.  The concatenated mini-batch $\hat{s}^n =  [s^n, \tilde{s}^n ]$ is utilized for    style-based learning and updating the model.  This  process not only mitigates the class-imbalance issue in each client but also provides a good platform for  style exploration by oversampling the  features. In our work, we reported the results with oversampling size of $B$ (which is  equal to the mini-batch size), while  the effect of the oversampling size is also reported in Fig. 4 of the main manuscript: A larger oversampling size leads to a better performance, and more importantly, our StableFDG outperforms the baseline even without any oversampling.
>
>
> **Class-imbalanced dataset:** The main reason of conducting feature-level oversampling is to generate new style statistics for our style exploration process. At the same time, as described in our answer to the previous question above, our feature-level oversampling mitigates the class-imbalance issue in each client, for class-imbalanced datasets.
>  The table below  shows the results on Office-Home dataset, where the class distribution is highly imbalanced in a multi-domain data distribution scenario (this result was already given in Section E.4 of the supplementary material). It can be seen that our class-balanced oversampling achieves higher performance over the simple random   oversampling, which validates the efficacy of mitigating the class imbalance problem in FL clients.
>
> * Effect of feature-level oversampling:
> |Methods             | Art | Cartoon | Photo | Sketch | Avg |
> | :---------------- | :------: | :------: |:------: |:------: |:------: |
> | Random oversampling    |    56.21| 54.43| 69.37| 72.29 | 63.08
> | Class-balanced oversampling   |   57.20 |  53.15 | 71.2 | 74.23 | **64.10**
>
>
>
>
> Again, thank you for your  time and efforts for reviewing our paper, and providing insightful comments. If there are any more clarifications we could provide, we would be grateful if you could let us know.

---

> > ### Comment · Reviewer_hWZZ · 2023-08-15
> >
> > I appreciate the answers from the authors, they address all my concerns.

---

> > > ### Author Response · Authors · 2023-08-17
> > >
> > > Dear Reviewer hWZZ
> > >
> > > Thank you for your time and efforts, and we are happy to hear that we have addressed all your concerns.
> > >
> > > Best, Authors of Paper 4257

---

### Official Review · Reviewer_wPUB · 2023-07-01

**Soundness:** 2 fair
**Presentation:** 2 fair
**Contribution:** 3 good
**Rating:** 5
**Confidence:** 3

**Summary:**

The authors observe that existing domain generalization (DG) algorithms face the issue in the setting of federated learning (FL) because of the lack of samples in each client's local datasets/domains. They propose StableFDG, which utilizes both style-based learning which allows the clients to explore styles different from that of the original source domain, and an attention-based feature highlighter, which captures the similarities and common characteristics of features of samples of the same class. The authors conduct their evaluation on datasets such as Office-Home and Digits-DG.

**Strengths:**

+ The evaluation is thorough, considering most of the popular benchmarks including the more challenging ones such as DomainNet.
+ The visuals are helpful and illustrative for readers to understand the proposed method

**Weaknesses:**

- It would seem that the work could greatly benefit from works in the field of style transfer, which also involves extracting domain knowledge. It would be better for the authors to remark on the connections between their style sharing and exploration with style transfer.

- It is not clearly explained what would happen after the style information is obtained by Equation 3. It's evident that this piece of information is used for style-shifting, etc., but how does such an operation relate to the intuition/big picture behind the approach?

**Questions:**

+ Have you considered style transfer approaches to boost your current style-based learning? Style transfer approaches also strive to extract the domain representation (style); would this field help or hinder the performance of StableFDG?

**Limitations:**

- The authors argue that, because the label information is not included in the style parameters, the proposed approach doesn't have privacy issues. However, style statistics/parameters, derived from the domains, still contain domain knowledge. It's not clear and not elaborated why the privacy issue is resolved simply because the labels aren't shared.

---

> ### Author Rebuttal · Authors · 2023-08-08
>
> We appreciate the reviewer for the positive comments and feedback. We are also thankful for the suggestions to  discuss  connections  between our style-based learning and existing style transfer. Our responses to the comments raised by the reviewer are given below.
>
>
> ### **Intuition behind equation 3**
>
> In a high-level, the style information (in equation 3) of other clients provide an opportunity for each client to train the model with the style information that is not in its local dataset. Fig. 2b in our manuscript provides insights into this. Without other clients' style information, a specific client will have only the styles within the orange region of Fig. 2b, which represents the styles in the client's local dataset. However, by utilizing other client's style information in equation 3 (blue region in Fig. 2b), the client is now able to expand the region in the style space to the blue region, by conducting style shifting to the blue area. As a result, the client's model is able to explore a wider style space when conducting style exploration, which leads to a better generalization. More detailed descriptions on how the parameters in equation 3 are utilized, are given in Lines 171-174 of our main manuscript.
>
>
> ### **Connections with style transfer literature**
>
> Related to the style transfer literature, our style-based learning can be  actually viewed as  a scheme that conducts  style transfer to the new styles (e.g., to the red points in Fig. 2b) based on the style statistics $\mu$ and $\sigma$, via Adaptive Instance Normalization (AdaIN) [ICCV'17].  Compared to  the style transfer methods that require a style transfer model for generating new images, AdaIN can conduct style transfer at feature-level during forward propagation in real-time, inspired by the instance normalization. Our style-based learning is built upon AdaIN as it  is especially beneficial  in FL settings as clients can perform style transfer without requiring an additional style-transfer model; the clients can directly conduct style transfer at feature-level during forward propagation (without generating images itself), using the style statistics via instance normalization.
>
> However, as can be seen from prior style transfer works [ICCV'21],  AdaIN may not achieve the best style transfer quality compared to  the style transfer methods that use an additional model, which is the cost for achieving lower computation and lower latency. Hence, we believe that if the FL clients can have large computational powers and storage spaces to conduct style transfer  using an additional model (and thus can possibly improve the quality of style transfer), and if a well-trained style transfer network is given, the achievable accuracy   could get improved with the cost of increased computation and delay. However, how our style-based learning should be designed/modified tailored to this new strategy, is currently open.
>
> Finally, we would like to note that the authors of [WACV'23] propose to directly generate new images at clients using a style-transfer network. However, as can be seen in Section B of our Supplementary Material, this scheme requires more computation time for processing more images, and more importantly, it achieves lower performance as the scheme is not able to explore enough styles (please refer to Table 1 and 2 in our main manuscript comparing CCST vs. StableFDG). Our style-based learning handles these issues by conducting style transfer via AdaIN instead of using an additional style-transfer network, and by exploring a wider style space via proposed style exploration.
>
> [ICCV'17]  Arbitrary style transfer in real-time with adaptive instance normalization
>
> [ICCV'21]  Domain-Aware Universal Style Transfer
>
> [WACV'23] Federated domain generalization for image recognition via cross-client style transfer
>
>
> ### **Domain knowledge and privacy**
>
> We appreciate the reviewer for pointing this out, which is correct: Although the  style parameters do not contain label information, they contain domain knowledge. Our original  intention was that   label information could be critical for privacy leakage as they directly   provide the third party to know  the classes of their images (e.g., types of diseases, types of objects in the image), while the   domain knowledge causes less privacy issue (e.g., weather information of the image, style information  of the image such as sketch, photo). However, we agree that there could be some applications where these domain information can incur privacy issue. In such applications, StableFDG  can be  conducted without  style sharing and shifting; each client directly performs style exploration using its local dataset, while skipping style sharing and shifting processes. In the table below, we show the performance of our StableFDG with and without style sharing/shifting, and compare them with the baselines: Although  the performance of StableFDG is slightly degraded when style sharing/shifting are not conducted,  it still outperforms all the baselines, demonstrating its advantage in applications where domain knowledge contain private information.
>
> * StableFDG with and without style sharing on PACS:
> |Methods             | Art | Cartoon | Photo | Sketch | Avg |
> | :---- | :--: | :----: |:---: |:---: |:--: |
> | MixStyle  |    80.07 | 	77.53 | 	96.23 | 	67.40 | 	80.31
> | DSU    |   80.53 | 	76.30 | 	95.37 | 	70.93 | 	80.78
> | StableFDG (w/o style sharing)    |  83.53  |  	78.53  |  	96.30	  |  75.27	  |  83.41
> | StableFDG (with style sharing)   | 83.97	| 79.10| 	96.27| 	75.67| 	83.75
>
> Overall, we agree with the reviewer’s comments and made efforts to discuss them in our response. Specifically, we tried to connect our work with other style transfer works, and provided discussions on how StableFDG should be modified domain knowledge contain private information. We hope to know if you are satisfied with our response. We would be happy to answer any remaining concerns you might have.

---

> > ### Author Response · Authors · 2023-08-20
> >
> > Dear Reviewer wPUB
> >
> > We appreciate the reviewer for the time and efforts for reviewing our paper. We have carefully considered your comments and tried to address them, especially the one on connecting our work with style transfer literature, and the one on domain knowledge and privacy. We are happy to answer any other concerns you might have during the remaining Author-Reviewer discussion period, which ends within 17 hours.
> >
> > Best, Authors of Paper 4257

---

### Official Review · Reviewer_fAoj · 2023-07-04

**Soundness:** 3 good
**Presentation:** 3 good
**Contribution:** 2 fair
**Rating:** 5
**Confidence:** 4

**Summary:**

In this paper, the authors present StableFDG, where a style-based strategy enables the model to reach a variety of new styles beyond each client's source domain, and an attention-based approach captures and emphasizes the important/common features of each category.

**Strengths:**

The StableFDG method proposed in the article has achieved good experimental results.
The experimental details in the article are particularly well articulated and richly illustrated.
The essay writing is well-organized and readable.

**Weaknesses:**

The idea of enabling the model to reach a variety of new styles beyond the source domain of each client has been proposed by other method [1], and what the authors have done is more of an incremental improvement.
In addition, the authors in a related work evaluate [1] to improve the computational and memory costs, but it itself takes up more memory than [1].
 [1] Junming Chen, Meirui Jiang, Qi Dou, and Qifeng Chen. Federated domain generalization for image recognition via cross-client style transfer. In Proceedings of the IEEE/CVF Winter Conference on Applications of Computer Vision, pages 361–370, 2023.

The method proposed in section 3.1 of the article is more like a combination of the methods mentioned in the authors' text.

In the "Where to apply style-based modules" subsection, the authors directly follow the way other methods apply modules, where more detailed ablation experiments could be done or the benefits or reasons for such application could be explained in the article.

The authors propose a new experimental setup, are the other methods in the paper based on this experimental setup, and what are the advantages of such a new setup over the previous ones?




**Questions:**

See the weakness.

**Limitations:**

The authors could have elaborated more fully on their new experimental setup.

---

> ### Author Rebuttal · Authors · 2023-08-08
>
> We appreciate the reviewer’s constructive suggestions. We make all of the raised points clearer below, with additional experiments as needed.
> ### **Differentiation from [1] and novelty of Sec. 3.1**
> Our style-based learning methodology in Sec 3.1 has three key components: (i) style sharing, (ii) selective style shifting, (iii) style exploration. [1] only shares some common idea with our first component (i.e., style sharing) in that each client shares some style information with others. Even in this component, the approaches employed in the two methods are widely different. We elaborate on three key points of differentiation from [1], and the novelty of Sec. 3.1 more generally, below.
>
> **(1) Style observation limitations in [1]:** It is important to clarify that [1] does not allow the model to observe *the styles that do not exist in the concatenated datasets of all FL clients*. To see this, consider a toy example with 2 clients, where the first client has style-information of $\mu_1$ and the second client has $\mu_2$ in its local dataset. [1] enables both clients to observe $\mu_1$ and $\mu_2$ (by sharing styles), but does not enable them to observe other styles beyond $\mu_1$ and $\mu_2$, limiting the style variability during training. StableFDG tackles this issue via style exploration combined with selective style shifting, which are the new perspectives in DG proposed by our work. As shown in Fig. 2b of our paper, style exploration exposes the model to various *novel styles that do not exist in the overall original dataset of the system*. This provides StableFDG with the following two key innovations: (a) by exposing the model to various novel styles beyond $\mu_1$ and $\mu_2$, the domain diversity of the source domains increases; (b) the model has better chances to cover diverse target domains during training since it can *explore a wider region in the style-space* (e.g., $\mu= \mu_1+\alpha(\mu_1-\mu_2$) via extrapolation). This leads to a significant performance improvement obtained by StableFDG (e.g. 8.57% gain on PACS in Table 1 of the manuscript).
>
> **(2) Computation/storage requirements:** [1] incurs more computation/storage compared to StableFDG because [1] directly generates *new data samples* in each client. On the other hand, our style sharing requires a much lower cost as it does not require generating the image itself. *Note that in Table 2 of the supplementary material, the first column indicates communication cost, not storage.* This very small 0.45\% additional communication burden is the fair cost for achieving significantly improved performance (4.75% gain over [1]) with reduced computation time.
>
> **(3) Attention-based feature highlighter:** Beyond our style-based learning in Sec. 3.1, the proposed attention-based feature highlighter (in Sec. 3.2) is another central contribution of our work. This handles the fundamental issues faced by DG in data-poor FL scenarios, by capturing the important/common characteristics of the samples in the same class. We stress that this approach has not been considered before, and we are one of the first to show that cross-attention can be strategically utilized to tackle the DG problem.
> ### **Where to apply the style module**
> **(1) Applied in layers where the style information is preserved:** Style-based learning is applied only in the 1st, 2nd, 3rd blocks among 4 residual blocks in ResNet-18. Note that at the output of the 4th block, label information is dominant rather than style information, which results in degraded performance when style-based schemes are applied. This is confirmed by our new experiments in **Table R4** above.
>
> **(2) Applied probabilistically:** Given the 1st, 2nd, 3rd block layers, style-based learning is applied probabilistically at one of the layers: It could be operated at the 1st, 2nd, 3rd, or none of the layers. This allows the model to see various combinations of style shift/exploration compared to the case where it is always applied at a fixed layer or none of the layers. This is confirmed by our new experiments in **Table R5** above.
> ### **New experimental setup**
> We first highlight that **experimental results in the conventional setup have been given in Section C of Supplementary Material**, to provide a fair comparison with the baselines [1], [27] in their settings. The results confirm the advantage of StableFDG in the same setting as prior works.
>
> Prior works [1], [27] considered a setup with N=3 clients, assuming that a single source domain is allocated to each client. We will represent this setup as (N=3, single). In our manuscript, we considered a setup with N=30, and considered both single-domain and multi-domain data distribution setups, e.g., (N=30, single) and (N=30, multi), which come from the following two motivations: (i) The original setup with only 3 clients does not well-reflect the practical FL scenario with a large number clients. This motivated us to consider more clients (N=30). (ii) The original setup does not consider each client having multiple domains in its local dataset. This motivated us to additionally consider the multi-domain setup.
>
> Overall, given the results with the original setting (in supplementary material), we believe that providing results with the new setup should be seen not as a weakness but as a strength of our presentation, because the results (i) further strengthen the validity of our method and (ii) reflect practical scenarios that are tailored to large-scale, heterogeneous FL settings.
>
> Again, thank you for your time and efforts. Your raised concerns made us think carefully, and we feel we have managed to clarify all the issues raised: we clarified the contributions of Sec. 3.1 compared to [1], provided intuitions/ablations on where to apply of style modules, and clarified our experimental setup. We hope that these are sufficient grounds for you to reconsider your rating. We would appreciate further opportunities to answer any remaining concerns you might have.

---

> ### Comment · Reviewer_fAoj · 2023-08-10
>
> The authors actively responded to the questions I asked and answered my concerns, the query about the new experimental setup was meant to make the meaning of the new setup clearer to the reader and hopefully the authors will add it to subsequent editions, my final score changes to Borderline Accept.

---

> > ### Author Response · Authors · 2023-08-11
> >
> > Dear Reviewer fAoj
> >
> > Thank you very much for your quick response and willingness to raise your score. We will make all points clearer based on our response, including the motivation of the new experimental setup.
> >
> > Earlier today, the Openreview system had not yet enabled reviewers to adjust their scores in the system. We noticed that has been fixed now, so if you could adjust your score accordingly we would appreciate it.
> >
> > If there is any additional clarification we could provide that would further increase your favorability of our paper, we would be grateful for any such opportunity.
> >
> > Best, Authors of Paper 4257

---

### Official Review · Reviewer_i1eF · 2023-07-06

**Soundness:** 3 good
**Presentation:** 3 good
**Contribution:** 3 good
**Rating:** 6
**Confidence:** 4

**Summary:**

This paper presents a novel domain generation method in the FL framework. The authors make two key contributions to enhance domain generalization in FL. Firstly, they introduce style-based learning, which empowers individual clients to explore new styles by leveraging both their own local statistics and those of other clients. This approach promotes domain diversity. Secondly, the authors propose an attention-based feature highlighter that captures similarities among the features of data samples within the same class. This component enables better learning of domain-invariant characteristics. To validate the efficacy of their proposed method, extensive experiments were conducted. The results demonstrate the superiority of the approach compared to existing methods on various benchmark datasets.

**Strengths:**

This paper addresses an important practical problem within the FL setup. It provides comprehensive coverage of related works and is well-written for easy understanding. The extensive experiments conducted effectively demonstrate the effectiveness of the proposed method. The supplementary material is valuable.

**Weaknesses:**

The adaptation of mixing styles and exploration to FL frameworks is straightforward. Also, the attention-based feature highlighter can be viewed as a variation of cross-attention.

**Questions:**

1) In this paper, statistics are recalculated at each round. Instead, could the running mean and variance of Batch Normalization (BN) be used as a substitute? Also, I am curious about the relationship between style augmentation and BN.
2) The attention-based feature highlighter combines self-attention and cross-attention for feature refinement. However, it remains unclear why this specific method is tailored to Federated Domain Generalization (DG). Further explanation is needed to clarify this aspect.
3) During training, both self-attention and cross-attention are used in Eq. (7), but during testing, only self-attention is employed. It would be beneficial to confirm if using self-attention alone during training is not sufficient.

**Limitations:**

Due to page constraints, some answers to certain questions were provided in the supplementary material, particularly in Supplementary Appendix E. It would be beneficial to include relevant content from the supplementary material into the main body of the paper.

---

> ### Author Rebuttal · Authors · 2023-08-09
>
> We appreciate the reviewer for the time and efforts, and providing helpful comments. Our responses are given below.
> ### **Style-based learning**
> We would like to first clearly differentiate the concept of existing style-augmentation [ICLR'21], [ICLR'22] and our style-based learning, and show that the proposed approach is based on key innovations that have not been considered in prior works, making the solution not straightforward. As shown in Fig. 2a of our paper, prior style-augmentation DG methods are constrained to generate new styles only near the original source domains, which leads to performance degradation especially in data-poor FL setups.
>
> We overcome these limitations by taking a different angle of attack: we intentionally and strategically expose the model to a wider style-space to effectively tackle DG in data-poor FL setups (as shown in Fig. 2b in the paper). The key advantages are: (i) the generated styles become more diverse beyond the styles each client has, and (ii) the generated styles can have *better chances to cover more diverse target domains* during training. We achieve this by developing our key components: (i) feature-level oversampling (that generates samples that will explore new styles) and (ii) parameter $\alpha$ that controls the exploration level. This is a new perspective in DG/FL proposed by our work to tackle the limitations of prior works, and has not been considered in the literature. Extensive experiments in 5 benchmark datasets also confirm the advantage of our approach.
> ### **Cross attention in StableFDG**
> Compared to existing works adopting attention, we would like to maintain that our paper provides a new direction by showing that *attention can indeed capture the domain-invariant characteristics* to tackle DG, in data-poor FL setups. We had to tackle several research questions to develop an attention strategy tailored to DG: First, how should we utilize attention to effectively capture important characteristics within each class, regardless of domains? We came up with the solution of strategically mixing cross and self attention, which enables the model to learn important characteristics across images (via cross-attention), and within the image (via self-attention). The next challenge was on how to design an inference strategy. Cross-attention was not applicable during inference because it is challenging to choose two samples from the same class for cross-attention without label information. Hence, we adopted self-attention during inference, which still provides StableFDG with significant performance advantages because our trained attention module is already able to capture key characteristics within the image.
>
> Moreover, as shown in our new experiments in **Table R1** above, adopting self attention solely during training is not the best option. This indicates that *different samples in the same class indeed share some common characteristics regardless of domains*, confirming the advantage of using both cross and self attention for training.
>
> Overall, we feel that (i) the idea of capturing domain-invariant characteristics in DG via attention, (ii) our training strategy (mixed cross, self-attention) and inference strategy (self-attention), and (iii) superior performance validated via experiments deserve merits. We would be happy to address any remaining concerns the reviewer might have.
>
> ### **Comparison with BN**
> We appreciate the comment. It has been revealed in recent studies [ICCV’17], [ICLR'21] that per-instance feature statistics (as in Instance Normalization [CVPR'17]) can represent style information of an image. Style augmentation schemes utilize these per-instance feature statistics to generate new styles to tackle DG. On the other hand, while the batch axis is also considered, per-batch feature statistics are obtained in BN. After normalizing the features using the comprehensive feature statistics, the BN layer shifts the features of all samples (instances) within a batch to the same distribution (according to learnable $\gamma$ and $\beta$). However, when the data distributions at training and testing are different as in DG setups, the performance of BN is degraded due to mismatch in feature statistics. Overall, BN is not able to fully capture the style of each instance and is susceptible to domain shifts. Our new experiments in **Table R2** above demonstrate this.
>
> ### **Why our attention is tailored to FL+DG**
> Thanks for this comment. In FL, each device typically has a limited number of samples, and thus models are prone to overfitting to each local dataset during local updates. Our attention strategy effectively learns domain-invariant features in such data-poor FL setups while mitigating the risk of overfitting to irrelevant feature information unrelated to each class. This is achieved by extracting common characteristics of samples in the same class and emphasizing them, while removing unimportant parts (e.g. background noises) that cause overfitting. Table 3 in the paper shows that applying our attention scheme alone already outperforms baselines.
>
> The follow-up question that the reviewer might be interested in is: Instead of the FL setup we focused on, can attention provide benefits in the centralized DG setup? Our new experiments in **Table R3** indicate that the proposed attention-based learning indeed captures the domain-invariant characteristics of samples, while the scheme provides more benefits in the FL setup where each client is prone to overfitting due to lack of data.
>
> Again, we appreciate the reviewer for the helpful comments. Your comments are clear and to the point, and we tried to clarify all the issues you raised: how/why our approach has new components, discussions on BN with new experiments, why attention is especially beneficial in FL, discussions on cross and self-attention with new experiments. In case there are remaining questions/concerns, we hope to be able to have an opportunity to further answer them.

---

> > ### Comment · Reviewer_i1eF · 2023-08-16
> >
> > I appreciate your thorough rebuttals to my questions and those of other reviewers. I've carefully reviewed your explanations. While some concerns were resolved in your rebuttal, I still find a slight lack of novelty. As a result, I have adjusted my rating to a weak accept.

---

> > > ### Author Response · Authors · 2023-08-17
> > >
> > > Dear Reviewer i1eF
> > >
> > > We appreciate the reviewer for raising the score. We will revise the manuscript according to the reviewer’s helpful comments. We will also try to make the contribution clearer, and emphasize that our paper is the first work in the DG and FL+DG literature to simultaneously focus on style diversity (based on style-based learning) and domain-invariance (based on attention), taking the best of both worlds with new technical approaches. Again, thank you for your time and efforts, and taking a more positive position for our work.
> > >
> > > Best, Authors of Paper 4257

---

### Author Rebuttal · Authors · 2023-08-09

We appreciate all reviewers for providing constructive comments, which have greatly helped us to improve the paper.

Due to the limited content we can provide in each response, we would like to share additional experimental results that **Reviewer i1eF** and **Reviewer fAoj** suggested here. For the other reviewers (Reviewer wPUB \& Reviewer hWZZ), all results are provided in our response corresponding to each reviewer.

&nbsp;


### Tables \& References for **Reviewer i1eF**

* **Table R1:** Using cross + self attention together vs. using self attention alone (DomainNet dataset)
>| Methods    |  Clipart |  Inforgraph |  Painting |  Quickdraw |  Real | Sketch |  Avg|
| :---- | :--: | :----: |:---: |:---: |:---: |:---: |:--: |
| StableFDG (with self attention alone)  |  61.77 | **24.88** |  48.28 |  14.15 |  59.78  |  52.41 |  43.55
| StableFDG (with self + cross attention)  |  **62.58**|  24.12 | **52.23**|  **14.87** |  **60.60** |  **52.50** |**44.48**
>
>$\rightarrow$ It can be seen that considering cross attention together improves  the performance compared to the scheme that uses self-attention alone. This indicates that *different samples in the same class indeed share some common characteristics regardless of the domain*,  confirming  the advantage of adopting  cross-attention  to   learn domain-invariant features.  By taking advantage of both cross and self attention, StableFDG enables the model to extract and learn important characteristics across images (via cross-attention), and within the image (via self-attention).

* **Table R2:** Effects of sharing style-information vs. BN features in StableFDG (PACS dataset)
>|Methods             | Art | Cartoon | Photo | Sketch | Avg |
| :---- | :--: | :----: |:---: |:---: |:--: |
| StableFDG (sharing feature statistics of BN)  |  82.84|  78.16|  94.70|   72.43|  82.03
| StableFDG (sharing style-information $\Phi_n$)  |  **84.10**|  **78.57**|  **95.40**|  **72.73**|  **82.70**
>
>$\rightarrow$ The performance drops for *all domains* when using feature statistics of BN instead of the style information $\Phi_n$. Here, the BN feature statistics are only used during feature sharing for the baseline, while other parts remain the same for a fair comparison.

* **Table R3:** Effect of proposed attention-based feature highlighter in a centralized DG setup (PACS dataset)
>|Methods             | Art | Cartoon | Photo | Sketch | Avg |
| :---- | :--: | :----: |:---: |:---: |:--: |
| StableFDG (centralized setup, without attention)  | 84.15| 79.45| 96.21| 77.09|84.23
| StableFDG (centralized setup, with attention)  |  **85.02**|  **79.65**|   **96.38**|  **78.45**|  **84.88**
>
>$\rightarrow$ The results show that  attention still provides performance improvements in the centralized setup by learning domain-invariant features, although the gain is slightly lower than the gain in the FL setup as shown in Table 3 of the main manuscript. These results indicate that the proposed attention-based learning indeed captures the domain-invariant characteristics of samples, while the scheme provides more benefits in the FL setup where each client is prone to overfitting due to lack of data.


[ICLR'21] Zhou et al., ``Domain generalization with mixstyle,'' ICLR 2021.

[ICRL'22]  Li et al., ``Uncertainty modeling for out-of-distribution generalization,'' ICLR 2022.


[ICCV’17] Huang et al., “Arbitrary style transfer in real-time with adaptive instance normalization,” ICCV 2017.

[CVPR'17] Ulyanov et al., ``Improved texture networks: Maximizing quality and diversity in feed-forward stylization and texture synthesis,'' CVPR 2017.

&nbsp;

### Tables \& References for **Reviewer fAoj**

* **Table R4:** Ablation experiments on applying style-based learning at different layers (PACS dataset)
>|Methods             | Art | Cartoon | Photo | Sketch | Avg |
| :---- | :--: | :----: |:---: |:---: |:--: |
| Style exploration at 1st,  2nd,  3rd layers (main manuscript) |  84.10	  |  78.57	  |  95.40  |  72.73  | 82.70
| Style exploration at 1st, 2nd,  3rd,  4th layers    | 82.99|	78.54|	94.13	| 73.35 |	82.25
>
>$\rightarrow$ If we consider the 4th residual block to apply our style-based learning, the performance gets degraded. This result confirms the insight that style-based learning should be conducted at the earlier layers where style information is preserved.

* **Table R5:** Ablation experiments on applying style-based learning probabilistically (PACS dataset)
>|Methods             | Art | Cartoon | Photo | Sketch | Avg |
| :---- | :--: | :----: |:---: |:---: |:--: |
| Style exploration (none of the layers) |  79.30	 |  76.24	 |  92.87	 |  68.48 |79.22
| Style exploration (fixed first layer)   |  74.56| 	73.72| 	90.60	|   62.75   | 	75.41
|Style exploration (probabilistic, main manuscript)   | **84.10**	|**78.57**	|**95.40**|**72.73**|	**82.70**
>
>$\rightarrow$ It can be seen that applying style exploration probabilistically achieves the best performance. This confirms the intuition that the probabilistic approach will let the model to  see various combinations of style shift and exploration (including the original style of the image), compared to the case where style exploration is always applied at a fixed layer or none of the layers.

  [1] Chen et al., ``Federated domain generalization  for image recognition via cross-client style transfer,'' WACV 2023.

  [27]  Nguyen et al., ``Fedsr: A simple and effective domain generalization method for federated learning,'' NeurIPS 2022.

---

> ### Comment · Area_Chair_1641 · 2023-08-18
>
> Thank the authors for the rebuttal. PCs and I have reminded the reviewers to respond to the rebuttals as soon as possible. The final decision will depend on both the reviews and rebuttal.
>
> @Reviewers: This message is yet another reminder. Please try to respond to the rebuttal asap.
>
> --AC

---

### Decision · Program_Chairs · 2023-09-21

**Decision:**

Accept (poster)

**Comment:**

Four experts reviewed the paper, and all recommended Borderline/Weak Accept. Reviewer hWZZ was especially excited about the work for its technical novelty and superior results. The other reviewers were not impressed with the paper's technical novelty, but they liked the experiments and found the rebuttal convincing. Hence, the decision is to recommend the paper for acceptance. The authors are encouraged to incorporate the reviewers' suggestions and the rebuttal into the revision.